# A new metric for understanding hidden political influences from voting records

**Corrado Possieri[1]\*, Chiara Ravazzi[2], Fabrizio Dabbene[2], Giuseppe C. Calafiore[2,3]**

**1** Istituto di Analisi dei Sistemi ed Informatica "A. Ruberti", Consiglio Nazionale delle Ricerche (IASI-CNR), Roma, Italy, **2** Istituto di Elettronica e di Ingegneria dell'informazione e delle Telecomunicazioni, Consiglio Nazionale delle Ricerche (IEIIT-CNR), c/o Politecnico di Torino, Torino, Italy, **3** Dipartimento di Elettronica e Telecomunicazioni, Politecnico di Torino, Torino, Italy

\* corrado.possieri@iasi.cnr.it

**Data Availability Statement:** All the data used in this paper are available from the Kaggle database at the following link https://www.kaggle.com/cpossieri/italian-senate-xvii-legislature.

## Abstract

Inspired by the increasing attention of the scientific community towards the understanding of human relationships and actions in social sciences, in this paper we address the problem of inferring from voting data the hidden influence on individuals from competing ideology groups. As a case study, we present an analysis of the closeness of members of the Italian Senate to political parties during the XVII Legislature. The proposed approach is aimed at automatic extraction of the relevant information by disentangling the actual influences from noise, via a two step procedure. First, a sparse principal component projection is performed on the standardized voting data. Then, the projected data is combined with a generative mixture model, and an information theoretic measure, which we refer to as *Political Data-aNalytic Affinity* (Political DNA), is finally derived. We show that the definition of this new affinity measure, together with suitable visualization tools for displaying the results of analysis, allows a better understanding and interpretability of the relationships among political groups.

## Introduction

In the past decades, many efforts have been spent by the scientific community on the study of mathematical models for opinion formation in social and belief systems [1, 2]. Among these models, the Friedkin and Johnsen's (F&J) opinion dynamics model [3] has been experimentally validated for deliberative groups of small and medium size [4]. According to this model, the agents' opinions evolve as a convex combination of others' beliefs and an initial condition. In this sense, agents are not completely open-minded, being persistently driven by an individual attachment due, for example, to the influence from a specific ideology [5, 6]. The key ingredient for estimating this *stubbornness* and, consequently, for offering insights in efficient control strategies for steering social behaviors towards desired patterns, is the development of new technically sound tools, able to extract low-dimensional information from social data [7–10].

In this paper we address the problem of understanding, via machine learning techniques, the attachment of individuals to their own group, and the underlying influence from

**Funding:** The author(s) received no specific funding for this work.

**Competing interests:** The authors have declared that no competing interests exist.

competing ideology groups, by using observations of public voting data in politics. Although other types of information could be used in principle for exploring relationships in the political scenarios, such as speeches and interviews, the voting patterns are particularly meaningful for their impact in the society and can reveal important behaviors and trends [11, 12]. Further, several voting data sets are already publicly available, and the trend in developed countries is to make these data more and more widely accessible, see, for instance, roll call data in the US congress, https://www.govtrack.us, and in the European parliament, http://www.europarl.europa.eu/plenary/en/minutes.html. However, although the data is publicly available, the ability to extract relevant information from it may be a challenging task. Oftentimes, data is provided by governmental institutions under the form of *minutes*, that is textual documents from which the actual quantitative voting data is to be extracted via costly human intervention, or via ah-hoc text analytics software. Alternatively, data can be obtained, sometimes at a cost, from research, nonprofit, or commercial parties, see, e.g., https://voteview.com and https://www.votewatch.eu.

The present research focuses on the activity in the Italian Senate during the XVII Legislature. The Italian political scenario represents an interesting case of study due to its complexity compared to other foreign deliberative institutions, since it is composed of several parties. Our data source is the nonprofit organization Openpolis, which tracks information about representatives and senators in Italy, including votes, monitoring government daily events, and providing statistics on politicians' actions. In our analysis, we focus on data that is classified by OpenPolis as *key votes*, i.e., those votes that are publicly available and considered as the most important, both for the relevance of the subject matter and for the political value. We also acquired the nominal membership of each senator to her/his political group, which will be used as side information. Note that these votes constitute just a portion of the total ensemble of votes made by a senator, which also include, e.g., the ones made in secret ballots.

The main contribution of this paper is the introduction of a new quantitative metric for measuring the affinity between each representative and the existing political parties. This affinity index, which we shall refer to as the *Political Data-aNalytics Affinity* (Political DNA), summarizes the degree of fidelity to a party, as well as the influence from other parties. It can be equivalently interpreted as a quantitative indicator of the degree of "rebellion" to the discipline of the party of nominal membership. Intuitively, it can be expected that representatives belonging to the same political party will express approximately homogeneous votes on a given bill. However, when bills are analyzed in their totality, say over a few years, a structure emerges which links each representative not only to his/her nominal party but also to other parties, indicating that the political orientation of a representative is possibly influenced by many diverse political visions. In extreme cases, we may observe representatives who nominally belong to some party, but whose votes indicate stronger political affinities with other parties, see, e.g., some examples in Fig 3.

The proposed metric is based on an information-theoretic ground, by modeling the votes as outcomes of a mixture of random variables and reformulating the computation of Political DNA as an estimation of class-posterior probabilities. The combination of this new metric with sparse learning techniques allows us to also select the most relevant bills determining variances in the political positions. Moreover, we develop ad-hoc visualization tools allowing an easier and immediate understanding of the affinity relationships.

Preliminary results concerning the Political DNA of the Italian senate have been presented in Longo *et al.* [13]. The main differences between the results given in this paper and those in Longo *et al.* [13] is that we here perform a deeper analysis considering all the political groups that were active during the XVII Legislature, we propose novel visualization tools that were

not provided in Longo *et al.* [13], and we illustrate how the Political DNA can be employed for determining the reciprocal influence among political groups.

## Background and related literature

### Italian senate

In Italy the Parliament holds legislative power, i.e., the faculty of making new laws. According to the principle of full bicameralism, it is composed by two houses with identical powers and functions: the Chamber of Deputies and the Senate of the Republic. In particular, the Senate consists of 315 elected members, the former Presidents of the Republic and life senators, appointed by the President of the Republic "for outstanding merits in social, scientific, artistic and literary fields". The Senate is chaired by the President of the Senate, elected by the senators in a secret ballot. Internally, the senators are joined in political groups but senators are free to migrate from the original political group to another during the Legislature.

Fig 1 shows the list of the political groups active at the end of the XVII Legislature (from the 15th of March 2013 to the 22nd of March 2018). If a political group contains multiple parties, only the most significant will be indicated. After elections, political groups form coalitions in order to create a multi-party pro-government majority on one side, and the opposition group on the other side. In the same table the political orientation (Left, Right, Center, Center-Right and Center-Left) and the role in the Legislature (Government-Opposition) of political groups are shown.

### Analysis of voting records

The analysis of voting records has a long history in social and political science. Several approaches have been proposed in the literature for scoring the political ideology from voting data [14, 15]. The most popular techniques belong to the family of Multidimensional scaling, such as Nominate, W-Nominate, and DW-Nominate [15, 16]. These methods are mainly used to produce graphical interpretations of political positions, representing high-dimensional data in a space with a lower dimension. Jenkins [17] used these methods to extract ideal points used as features in estimating party influence and to determine the ideological rank order in the US Congress [8].

Although these techniques are based on empirically-grounded models of opinion formation and can be interpreted both in probabilistic and in geometric framework, the algorithms are highly computationally intensive. Alternative to previous methods, Principal Component Analysis allows extraction of the relevant information from highly correlated data by reducing the number of dimensions while retaining most of the variance of the data, and it provides a compact and meaningful representation of the original data. PCA is commonly used for the Ideology Analysis of legislators in the US Congress. In particular, several data set are used for this analysis, including political orientation (Left-Right), co-sponsorship, committees seats, campaign contributions, to mention just a few. The Italian Senate, however, presents some intrinsic differences with respect to the US Congress, making the analysis more challenging and calling for new tools. In particular, we emphasize the higher number of legislators (although smaller than the one of the Chamber of Deputies), their high fragmentation among several political groups, several ideologies/coalitions/political orientations, and high migration between parties [18].

## Methods and data

### Notation

Natural and real numbers are denoted by $\mathbb{N}$ and $\mathbb{R}$, respectively. We denote column vectors with lower-case letters, and matrices with capital letters. Given a matrix $X \in \mathbb{R}^{m \times n}$, $X^\top$ denotes

| Group | Symbol | Color | Parties | Orientation | Role | Senators |
|-------|--------|-------|---------|-------------|------|----------|
| ALA | | | ALA-PRI | Center | Government | 13 |
| AUT | | | AUT(SVP-UV-PATT-UPT)-PSI | Center | Government | 19 |
| FdL | | | FL(Id-PL, PLI) | Center-Right | Opposition | 10 |
| GAL | | | GAL-UDCeDC | Center-Right | Opposition | 14 |
| LeU | | | Art.1-MDP-LeU | Left | Opposition | 16 |
| Lega | | | Lega | Center-Right | Opposition | 12 |
| M5S | | | M5S | Independent | Opposition | 35 |
| Misto | | | Misto | Mixed | Opposition | 27 |
| NCD | | | AP-CPE-NCD-NCI | Center-Right | Government | 24 |
| NcL | | | NcL | Center | Government | 11 |
| PD | | | PD | Center-Left | Government | 101 |
| PdL | | | FI-PdL | Center-Right | Opposition | 52 |

**Fig 1. Political groups active at the end of the XVII Legislature.** The acronyms stand for: ALA: Alleanza Liberalpopolare-Autonomie; AUT: Per le Autonomie; FdL: Federazione delle Libertà; GAL: Grandi Autonomie e Libertà; LeU: Liberi e Uguali; Lega: Lega Nord e Autonomie; M5S: Movimento 5 Stelle; NCD: Nuovo Centrodestra; NcL: Noi con l'Italia; PD: Partito Democratico; PdL: Il Popolo delle Libertà.

its transpose, $X_{ij}$ is the entry corresponding to $i$-th row and $j$-th column. **1** denotes a (column) vector of all ones. We use the notation $x^{(i)} \in \mathbb{R}^n$ for the column vector corresponding to the $i$-th row, i.e.

$$X = \begin{bmatrix} x^{(1)\top} \\ \vdots \\ x^{(m)\top} \end{bmatrix}.$$

Given $z \in \mathbb{R}^n$, we denote the Euclidean norm and $\ell_0$-pseudonorm (number of non-zero elements) with $\|z\|_2$ and $\|z\|_0$, respectively. We denote the Frobenius norm of a matrix $X \in \mathbb{R}^{m \times n}$ by

$$\|X\|_F := \sqrt{\sum_{i=1}^{m} \sum_{j=1}^{n} X_{ij}^2}.$$

## Dataset

The dataset under consideration contains the final votes of members of the Italian Senate during the XVII Legislature. A field in the dataset specifies the membership of each senator to his/her most recent political group. The data consist in the vote of each senator for each proposed bill, expressed as "Favorevole" (for), "Contrario" (against), "Astenuto" (abstention) and "Assente" (not in chamber). These data have been cleaned by removing the senators who never voted and the bills voted in a secret ballot. After this cleaning, the dataset contains the votes of $m = 334$ senators on $n = 160$ bills. The dataset is available at the following link: https://www.kaggle.com/cpossieri/italian-senate-xvii-legislature.

## Encoding and preprocessing

The votes of senators have been encoded in a *vote matrix* $Z \in \{-1, 0, 1\}^{m \times n}$, where each row represents a senator and each column a bill. The $(i, j)$th entry $Z_{ij}$ of such a matrix equals $+1$ if the $i$th senator was favorable to the $j$th bill, $-1$ if he/she was either against the $j$th bill or abstained from voting, and $0$ in all the other cases (namely, if he/she was not in chamber). This encoding is coherent with the voting rule of Italian Senate during the XVII Legislature, according to which abstention is essentially equivalent to a rejection. This vote matrix has been standardized over the columns, as in [19]:

$$X_{ij} = \frac{Z_{ij} - \frac{1}{m}\sum_{i=1}^{m} Z_{ij}}{\sqrt{\sum_{i=1}^{m}\left(Z_{ij} - \frac{1}{m}\sum_{i=1}^{m} Z_{ij}\right)^2}}, \tag{1}$$

$i = 1, \ldots, m, j = 1, \ldots, n$, thus obtaining the centered and standardized matrix $X \in \mathbb{R}^{m \times n}$. Each vector $x^{(i)} \in \mathbb{R}^n$ of this matrix represents the centered and standardized votes of senator $i$, which is associated to a scalar index $\omega^{(i)} \in \{1, \ldots, r\}$ representing his/her most recent political group. Note that if we define $\bar{z}^{\top}$ as the average of the rows of $Z$, that is $\bar{z}^{\top} = \frac{1}{m}\sum_{i=1}^{m} z^{(i)\top}$, then $\bar{z}^{\top}$ has the interpretation of the row $n$-vector of votes from the average senator on the $n$ bills. Also, if we define $\bar{\sigma}^{\top}$ as the row $n$-vector containing the sample standard deviation along the columns of $Z$, that is $\sigma_j^2 = \frac{1}{m}\sum_{i=1}^{m}(Z_{ij} - \bar{z}_j)^2$, for $j = 1, \ldots, n$, then $\bar{\sigma}_j$ gives a measure of the variability of opinions of the senators on the $j$th bill. The standardized data matrix $X$ in (1) thus contains, in each column (bill) $j = 1, \ldots, n$, the votes of the senators, centered around the average vote $\bar{z}_j$ for that bill, and scaled according to the standard deviation of the votes for that same bill. In compact matrix notation, we can express $X$ as

$$X = (Z - \mathbf{1}\bar{z}^{\top})S^{-1}, \quad \text{where } S = \text{diag}(\bar{\sigma}_1, \ldots, \bar{\sigma}_n).$$

## Learning political DNA from raw data

The main objective of the proposed methodology is to infer, for each senator $i \in \{1, 2, \ldots, m\}$, his/her Political DNA, that is, a vector $\pi^{(i)} \in [0, 1]^r$ whose entries $\{\pi_y^{(i)}\}_{y \in \{1, \ldots, r\}}$ represent the

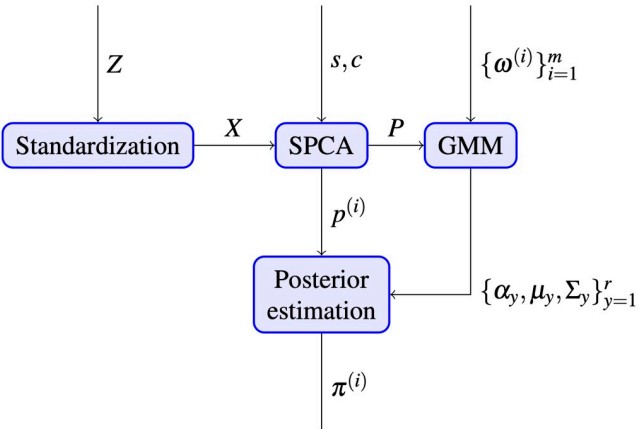

**Fig 2. Procedure for the extraction of Political DNA.**

influence exerted from group $y$ on senator $i$, with the property that $\sum_{y=1}^{r} \pi_y^{(i)} = 1$. In this section, we briefly describe the technique that has been used to achieve this objective. In particular, we assume a Gaussian mixture generative model for representing the voting pattern of senators, and then use Bayes' rule to extract the political DNA. The raw data are firstly encoded in a vote matrix, which is centered and standardized as in (1). We then apply a dimensionality reduction technique (namely, Sparse Principal Component Analysis, SPCA), thus obtaining lower dimensional data. This first prefiltering step allows for reduction of noise and extraction of the relevant features from the data matrix. These filtered data are then used to learn a Gaussian Mixture Model (GMM), from which the Political DNA is finally extracted via posterior probability computation. Fig 2 summarizes this procedure.

**Dimensionality reduction.**   Vote data matrices are typically noisy, containing possibly spurious information. A classical computational approach used to to better expose the information contained in the data matrix is the Principal Component Analysis (PCA), which is related to the singular value decomposition of $X$, in the dyadic form

$$X = \sum_{k=1}^{s} \sigma_k u_k v_k^{\top}, \tag{2}$$

where $s$ is the rank of $X$, $\sigma_1 \geq \sigma_2 \cdots \geq \sigma_s \geq 0$ are the singular values, and $u_k \in \mathbb{R}^m$, $v_k \in \mathbb{R}^n$, $k = 1, \ldots, s$, are the left and right singular vectors of $X$, respectively, which also form orthonormal bases for $\mathbb{R}^m$ and $\mathbb{R}^n$, respectively. The well-known Eckart-Young-Mirsky theorem states that the truncation of (2) at the $q$th term (with $q \leq s$), namely $X_q = \sum_{k=1}^{q} \sigma_k u_k v_k^{\top}$, provides the best rank-$q$ approximation to $X$, in both the Frobenius and the spectral norm metrics. The $q$ pairs of vectors $(u_k, v_k)$, $k = 1, \ldots, q$, constitute the first $q$ so-called *principal components* of $X$. The relative approximation error, in the Frobenius norm, for $q < s$, is given by

$$e_q^2 = \frac{\|X - X_q\|_F^2}{\|X\|_F^2} = \frac{\sum_{k=q+1}^{s} \sigma_k^2}{\sigma_{\text{tot}}^2}, \tag{3}$$

where $\sigma_{\text{tot}}^2 \doteq \|X\|_F^2 = \sum_{k=1}^{s} \sigma_k^2$ is the so-called *total variance* of the data matrix. A small $e_q$ value signifies that the variability of the data matrix is well explained by means of its first $q$ principal components, who capture the most relevant spatial directions along which data variation occurs. Equivalently, along the residual directions $(u_k, v_k)$, $k = q + 1, \ldots, s$, there is little data

variability, which is attributed to noise, and thus these directions can be discarded. This standard PCA approach realizes effective dimensionality reduction whenever $e_q$ is small for $q$ values reasonably smaller than the original rank $q$. However, one problem with standard PCA is that when we look at, say, the first principal directions $(u_1, v_1)$, these vectors are typically dense (as opposed to sparse) vectors. If we were to approximate the vote matrix by its first principal components only, we would have

$$X \simeq X_1 = \sigma_1 u_1 v_1^\top,$$

where the entries in $u_1 \in \mathbb{R}^m$ represents senators' influence coefficients on the votes, and the entries in $v_1$ represent bills' influence on the vote outcomes. A sparse $u_1$ would highlight which senators are the leading actors in influencing the votes, and a sparse $v_1$ which bills are the most relevant for capturing the main voting trends. For pursuing both dimensionality reduction and interpretability, we thus employed a sparse version of the PCA, namely the Sparse-PCA (SPCA) method discussed in [20]. Computationally, given the centered and standardized matrix $X$, and letting $\Sigma = X^\top X$, the SPCA aims at determining a vector $v_1 \in \mathbb{R}^n$ that maximizes $v_1^\top \Sigma \, v_1$, subject to the constraints that $\|v_1\|_2 = 1$ and that the cardinality (i.e., the number of nonzero elements) of $v_1$ is no larger than a given positive integer $c \in \mathbb{N}$, that is

$$\left| \begin{aligned} \max \quad & v_1^\mathsf{T} \sum v_1, \\ & \|v_1\|_2 = 1, \\ & \|v_1\|_0 \leq c. \end{aligned} \right. \tag{4}$$

Due to the cardinality constraint, the problem given in (4) is NP-hard in the strong sense [21]. For this reason, several approaches have been proposed in the literature to approximately solve (4), including a regression framework [20], an approach based on semidefinite programming relaxation [22], and inverse and truncated power methods [23, 24]. The latter was used in this paper for performing SPCA on the standardized vote data matrix. Note that, once the vector $v_1$ has been computed, the SPCA algorithm can be applied again to a suitably "deflated" version of the original matrix, $\Sigma_1 = \Sigma - (v_1^\top \Sigma \, v_1) \, v_1 \, v_1^\top$, so to extract the second principal component $v_2$, and so on [22, 25, 26]. By stacking the vectors $v_1, \ldots, v_q$ so to obtain the matrix $V_q = \begin{bmatrix} v_1 & \cdots & v_q \end{bmatrix} \in \mathbb{R}^{n \times q}$, the vote matrix is projected onto the $q$-dimensional space $\mathbb{R}^q$, thus obtaining the projected data matrix $P = X V_q \in \mathbb{R}^{m \times q}$. Each element $p_{ij}$ of the $P$ matrix represents a linear combination of all the votes of the $i$th senator, with coefficients given by the $j$th principal direction $v_j$. Since $v_j$ is $c$-sparse, only the $c$ "most relevant" bills actually contribute to such linear combination.

**Gaussian mixture model.** We use a Gaussian Mixture Model (GMM) to represent the projected data $P$. Such a generative model assumes that there are $r$ classes (each class, in our context, actually corresponding to a political party), and that the projected vote vector, conditional on the class being $y$, is a Gaussian random variable with mean $\mu_y$ and covariance matrix $\Sigma_y$, $y = 1, \ldots, r$, where $r$ denotes the number of classes. In our analysis we considered $r = 12$ classes (political groups). By letting $\alpha_y$ represent the latent class probabilities, $\alpha_y \geq 0$, $\sum_{y=1}^r \alpha_y = 1$, the distribution $f(p)$ of the projected vote vector $p$ is a Gaussian mixture

$$f(p) = \sum_{y=1}^r \alpha_y \mathcal{N}(p | \mu_y, \Sigma_y), \tag{5}$$

where $\mathcal{N}(p|\mu_y, \Sigma_y)$ is a Gaussian density

$$\mathcal{N}(p|\mu_y, \Sigma_y) = \frac{\exp\left(-\frac{1}{2}(p - \mu_y)^\top \Sigma_y^{-1}(p - \mu_y)\right)}{\sqrt{(2\pi)^n \det(\Sigma_y)}}.$$

For each class, the mean and the covariance matrix are estimated by using the maximum likelihood principle [27]. In particular, defining the set $\mathcal{G}_y = \{i \in \{1, \ldots, m\} : \omega^{(i)} = y\}$, $y = 1, \ldots, r$, we obtain

$$\alpha_y = \frac{|\mathcal{G}_y|}{m}, \quad \mu_y = \frac{1}{|\mathcal{G}_y|}\sum_{i \in \mathcal{G}_y} p^{(i)}, \quad \Sigma_y = \frac{1}{|\mathcal{G}_y| - 1}\sum_{i \in \mathcal{G}_y}(p^{(i)} - \mu_y)(p^{(i)} - \mu_y)^\top,$$

where $|\mathcal{G}_y|$ denotes the cardinality of the $y$th class.

**Extraction of the DNA from data.** Once the parameters of the Gaussian mixture model (5) have been estimated as described above, we consider the latent class variable $z_y$, which is such that $z_y = 1$ if the datum $p$ comes from class $y$, and 0 otherwise, and define the conditional probability of the latent variable, given the observed datum $p$ (the projected votes vector). In formulas, using Bayes' rule, we have that

$$\pi_y \doteq \mathbb{P}(z_y = 1|p) = \frac{\mathbb{P}(z_y = 1)f(p|z_y = 1)}{f(p)} = \frac{\alpha_y \mathcal{N}(p|\mu_y, \Sigma_y)}{f(p)}$$

$$= \frac{\alpha_y \mathcal{N}(p|\mu_y, \Sigma_y)}{\sum_{v=1}^{r} \alpha_v \mathcal{N}(p|\mu_v, \Sigma_v)}.$$

We shall view $\alpha_y$ as the prior probability of $z_y = 1$, and $\pi_y$ as the corresponding posterior probability, once we have observed $p$. Also, $\pi_y$ can be interpreted as the influence that the group $y$ has in explaining the observation $p$. The Political DNA of each Senator $i = 1, \ldots, m$, is then defined as the vector

$$\pi^{(i)} = \begin{bmatrix} \pi_1^{(i)} & \cdots & \pi_r^{(i)} \end{bmatrix}^\top,$$

where

$$\pi_y^{(i)} = \frac{\alpha_y \mathcal{N}(p^{(i)}|\mu_y, \Sigma_y)}{\sum_{v=1}^{r} \alpha_v \mathcal{N}(^{(i)}|\mu_v, \Sigma_v)}, \quad y = 1, \ldots, r,$$

containing the influences from all the $r$ classes, upon evidence of the $i$th senator's projected votes vector $p^{(i)}$.

## Visualization of political data

In this section, we introduce three tools that we employed for producing easily interpretable visualizations of the political influence data obtained via the proposed extraction technique.

**Political affinity map.** The Political Affinity Map is a bi-dimensional representation of the Political DNA of each senator. More precisely, we draw a regular polytope whose vertexes $\{\gamma_y\}_{y=1, \ldots, r}$ represent the groups. The $i$th senator is then represented as a polytope with vertexes $\{\beta_y^{(i)}\}_{y=1,\ldots,r}$, where $\beta_y^{(i)} = \pi_y^{(i)}\gamma_y$. The resulting plot is a spider chart (usually referred to also as radar plot or Kiviat diagram) in which the length of each "spike" is proportional to the influence of the $y$th group to the $i$th senator and adjacent "spikes" are connected via a segment. Depending on the ordering of the political groups, this type of plots allows one to undercover

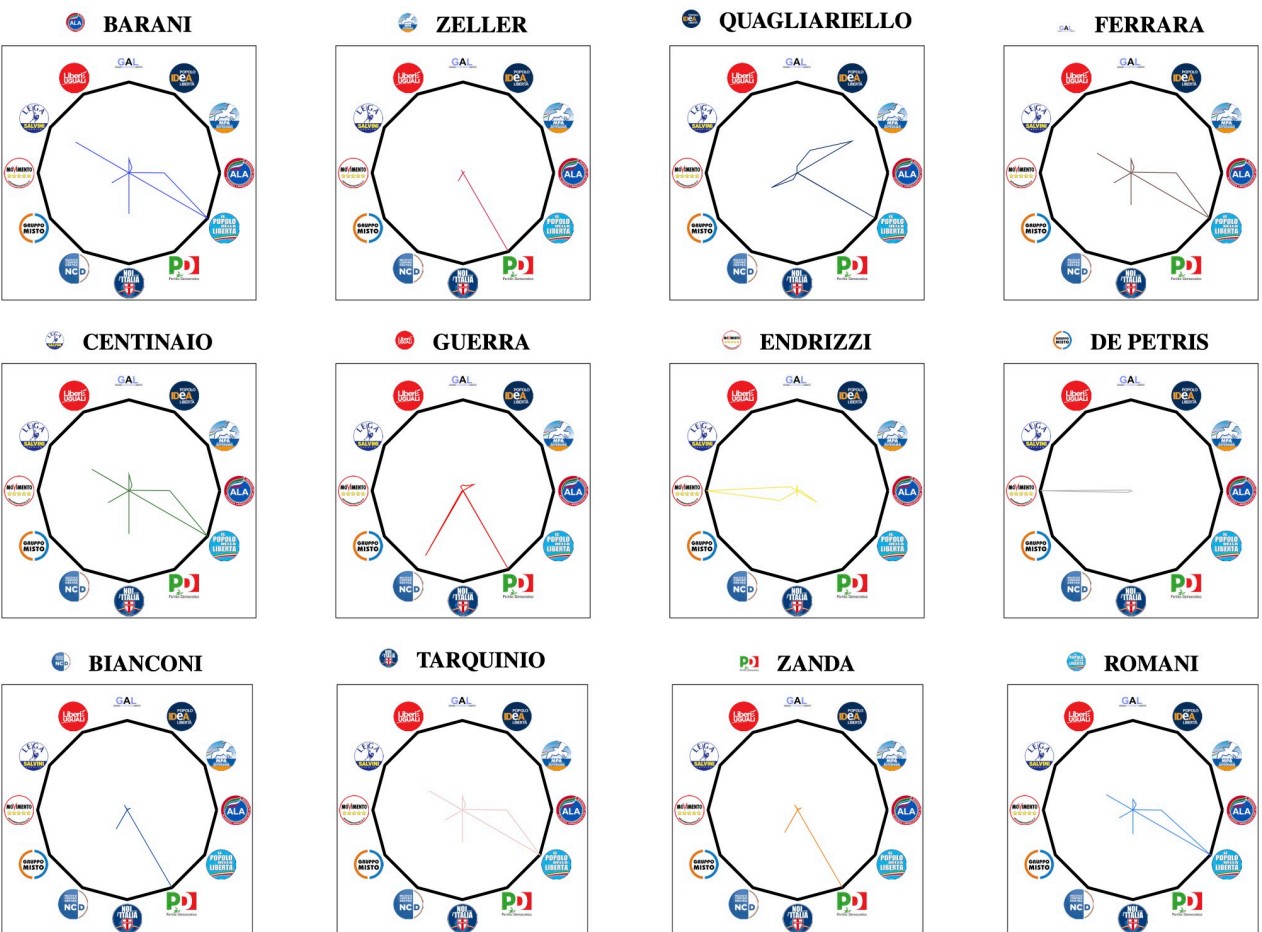

**Fig 3. Political affinity maps of the parliamentary leaders of each group obtained using the technique given in section with $q = 2$ and $c = 20$.**

the shift of each senator toward political parties different from the nominal one. For instance, larger areas of the polytope indicate that the ideology of the senator is influenced by several political parties, whereas unitary spikes in the direction of his/her nominal affiliation indicate that his/her ideology is consistent with the one of the party. Further, the Political Affinity Map allows one to determine senators with similar ideologies, political clusters, and the presence of outliers; see, e.g., [28].

Fig 3 depicts the Political Affinity Maps of the leaders of political groups in the Italian Senate, obtained using the proposed extraction technique, with $q = 2$ and $c = 20$.

**Segmentation plot.** The segmentation plot represents the Political DNA of each senator as a segmented bar. In particular, the Political DNA of the $i$th senator is represented as portions of the bar of different colors (each color corresponds to a political group) of length proportional to $\pi_y^{(i)}$. Fig 4 depicts the Segmentation Plots of the parliamentary leaders of each group obtained using the proposed DNA extraction technique with $q = 2$ and $c = 20$.

It is worth noting that the Political Affinity Map and the Segmentation Plot are just two different graphical representations of the Political DNA. These two representations allow to gather different information about the ideology of senators. For instance, the Political Affinity Map is useful for identifying clusters and detecting the presence of outliers. On the other hand, the Segmentation Plot is useful for undercovering the reciprocal influence among different

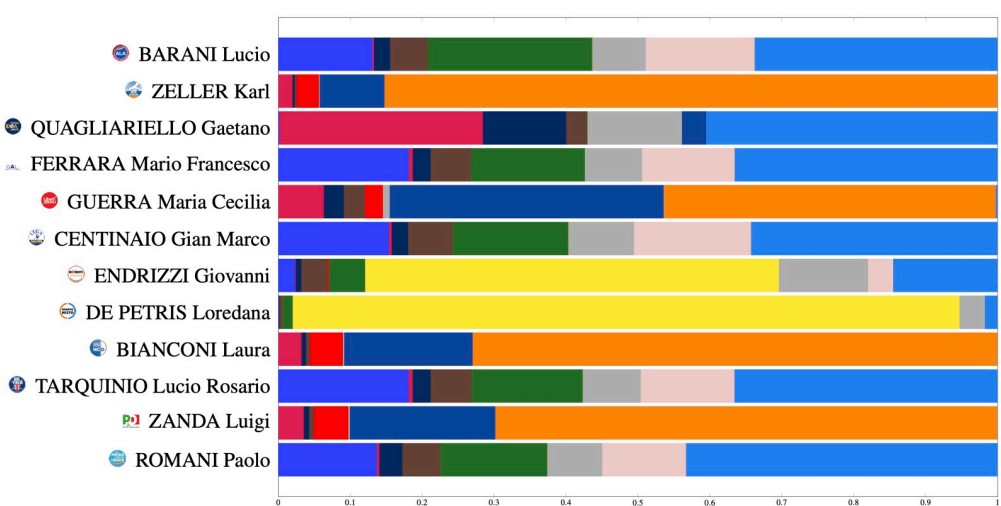

**Fig 4. Segmentation plots of the parliamentary leaders of each group obtained using the proposed DNA extraction technique with $q = 2$ and $c = 20$.**

parties and for identifying the leading groups. Therefore, the combined use of these two representation allows one to have a complete picture of the relationships among different parties.

**Reciprocal influence map.** The Political DNA can also be used for identifying the mutual influence among parties. Indeed, the average influence of the $y$th group to the $v$th group is given by

$$\Pi_y^v = \frac{1}{|\mathcal{G}_v|} \sum_{i \in \mathcal{G}_v} \pi_y^{(i)}.$$

In particular, the matrix $\Pi$, whose $(y, v)$th entry is $\Pi_y^v$, represents the average probability that a senator belonging to the $y$th group voted in a manner similar to a senator belonging to the $v$th group. We denote by Reciprocal Influence Map the graphical representation of such a matrix.

## Results and discussion

### Case study: Votes in the Italian senate during the XVII legislature

In this section, we present the results obtained by using the proposed technique on the vote data of the Italian Senate during the XVII Legislature. Selected Political Affinity Maps and Reciprocal Influence Maps obtained using the proposed DNA extraction technique with different values of the parameters $q$ and $c$ are shown in Figs 5 and 6. The nominal membership of each senator has been represented using the color code in Fig 1.

Although there is no an obvious ground truth to compare against, some considerations are in order. For each experiment, we computed the expressed variance (E-Var), defined as the ratio between the variance of the projected data and the one of the original data.

By looking at these numerical results, we may formulate the following observations:

- as expected, the E-Var of the data increases as a function of the number $q$ of principal components considered and of their cardinality level $c$;

- large values of $q$ and $c$ lead the senators' political positions to shift towards their nominal affiliation, which is represented in the Political Affinity Map as a "spike" of unitary length in the direction of their nominal affiliation. Therefore, noise filtering through decreasing the

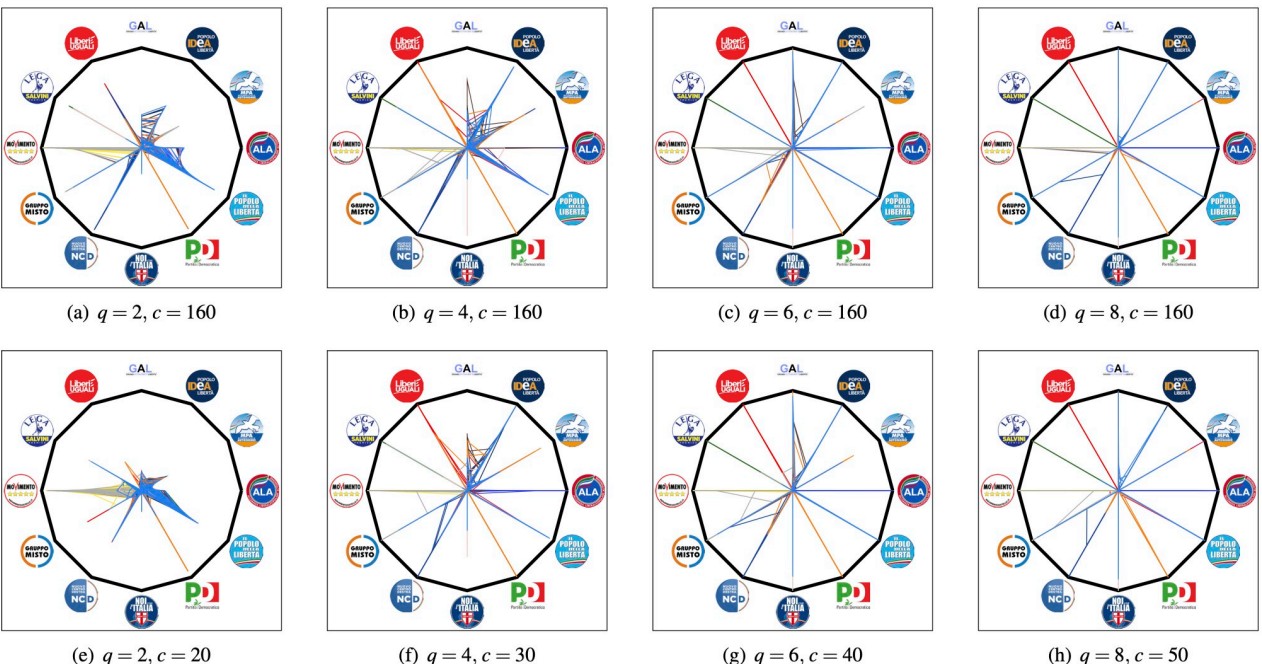

**Fig 5. Political affinity maps of all the senators for different values of *q* and *c* (*c* = 160 corresponds to full, non-sparse PCA).** The nominal membership of each senator is represented via the color code given in Table 1.

values of $q$ and $c$ reveals subtle structures of the political affinities among voters. It is worth noticing that this filtering property is the result of the use of SPCA for dimensionality reduction. In fact, subtle structures are usually confined in minor components [29–31], and filtering them out usually only maintains the global trends. On the other hand, SPCA allows one

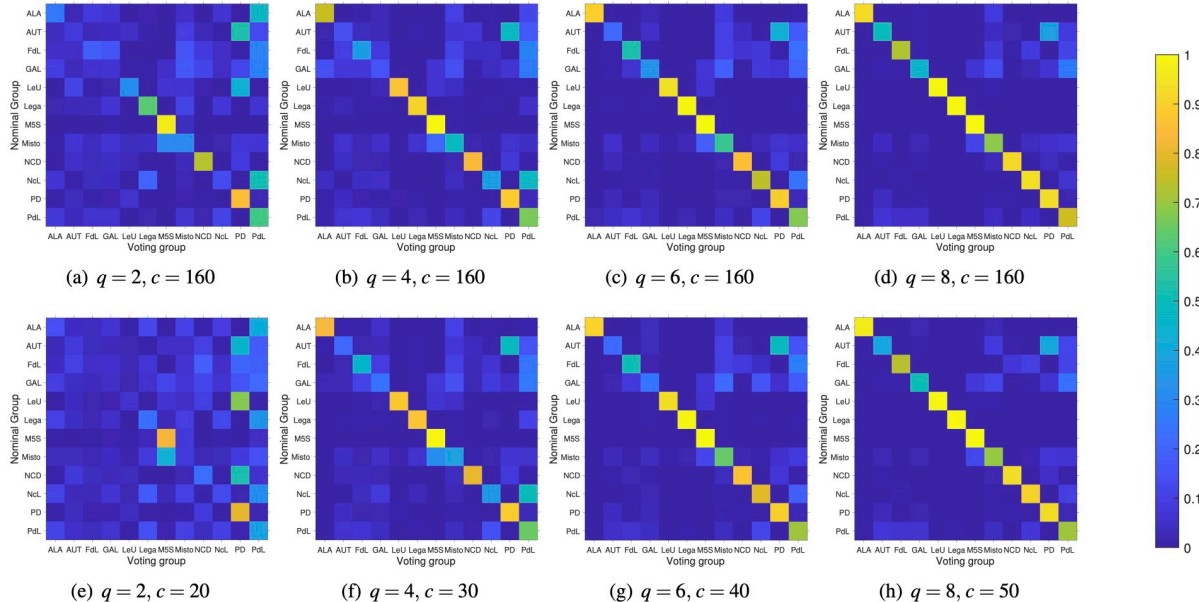

**Fig 6. Reciprocal influence maps for different values of *q* and *c*.** As shown by such plots, large values of $q$ and $c$ hide the reciprocal influence of each group on the others. Note that $c = 160$ corresponds to non-sparse PCA.

**Table 1. E-Var for different values of *q* and *c*.**

| | | $q$ | | | | | | | |
|---|---|---|---|---|---|---|---|---|---|
| | | 2 | 3 | 4 | 5 | 6 | 7 | 8 | 9 |
| $c$ | 20 | 14.247 | 18.889 | 20.485 | 23.433 | 26.663 | 27.735 | 28.57 | 29.315 |
| | 40 | 29.501 | 33.75 | 37.801 | 39.896 | 41.83 | 42.986 | 43.856 | 45.025 |
| | 60 | 40.111 | 44.78 | 47.963 | 50.166 | 52.214 | 53.461 | 54.716 | 55.628 |
| | 80 | 47.429 | 52.197 | 55.401 | 57.683 | 59.73 | 61.044 | 62.323 | 63.291 |
| | 100 | 52.405 | 56.945 | 60.238 | 62.534 | 64.61 | 65.967 | 67.265 | 68.287 |
| | 120 | 56.81 | 61.18 | 64.515 | 66.818 | 68.92 | 70.306 | 71.604 | 72.777 |
| | 140 | 58.999 | 63.367 | 66.72 | 69.024 | 71.137 | 72.532 | 73.842 | 75.010 |
| | 160 | 59.378 | 63.757 | 67.115 | 69.419 | 71.533 | 72.93 | 74.247 | 75.413 |

to focus just on the most relevant votes, hence undercovering hidden influences among individuals with different ideologies;

- similarly, large values of $q$ and $c$ tends to uniform the reciprocal influence of each group on the others;

- SPCA is useful also to identify the most significant bills influencing the projected vote vectors in the $q$-dimensional space. Table 2 reports the short description of the 10 most importantbills identified by SPCA, performed using $q = 10$ and $c = 1$.

Regarding the political structure of the Italian Senate, Fig 7 depicts the Segmentation Plots for selected values of the parameters $q$ and $c$, using the color coding given in Fig 1. By inspecting Fig 7 we observe the following facts:

- the groups "Misto," "GAL" and "AUT" are the first to spread out when projecting the data in a lower dimensional space. This is to be expected, since these groups collect senators of very different ideologies, and many senators "migrated" to this group after leaving their respective original groups. The Political DNA allows recovering their original membership or political orientation; see Fig 8.

- the group "M5S" is one of the most compact groups in the Senate, with senators remaining strongly cohesive even for low values of $q$ and high sparsity levels. This can be explained considering that a code of conduct—available at Codice Etico Movimento 5 Stelle—binds the

**Table 2. Most significant bills identified by SPCA.**

| Description |
|---|
| European delegation 2013 - DDL n. 587 - Final vote |
| Resignation of senator Mangili |
| Jurisdiction on ethical issues - DDL n. 1429 |
| Liability of magistrates - DDL n. 1070 - Final vote |
| Rosatellum bis - DDL n. 2941 - Final vote |
| Daily allowance for lifetime senators |
| Azzollini house Arrest |
| Anti femicide - DDL n. 1079 - Final vote |
| Health decree - DDL n. 298 |
| Public Debits - DDL n. 662 - Final vote |

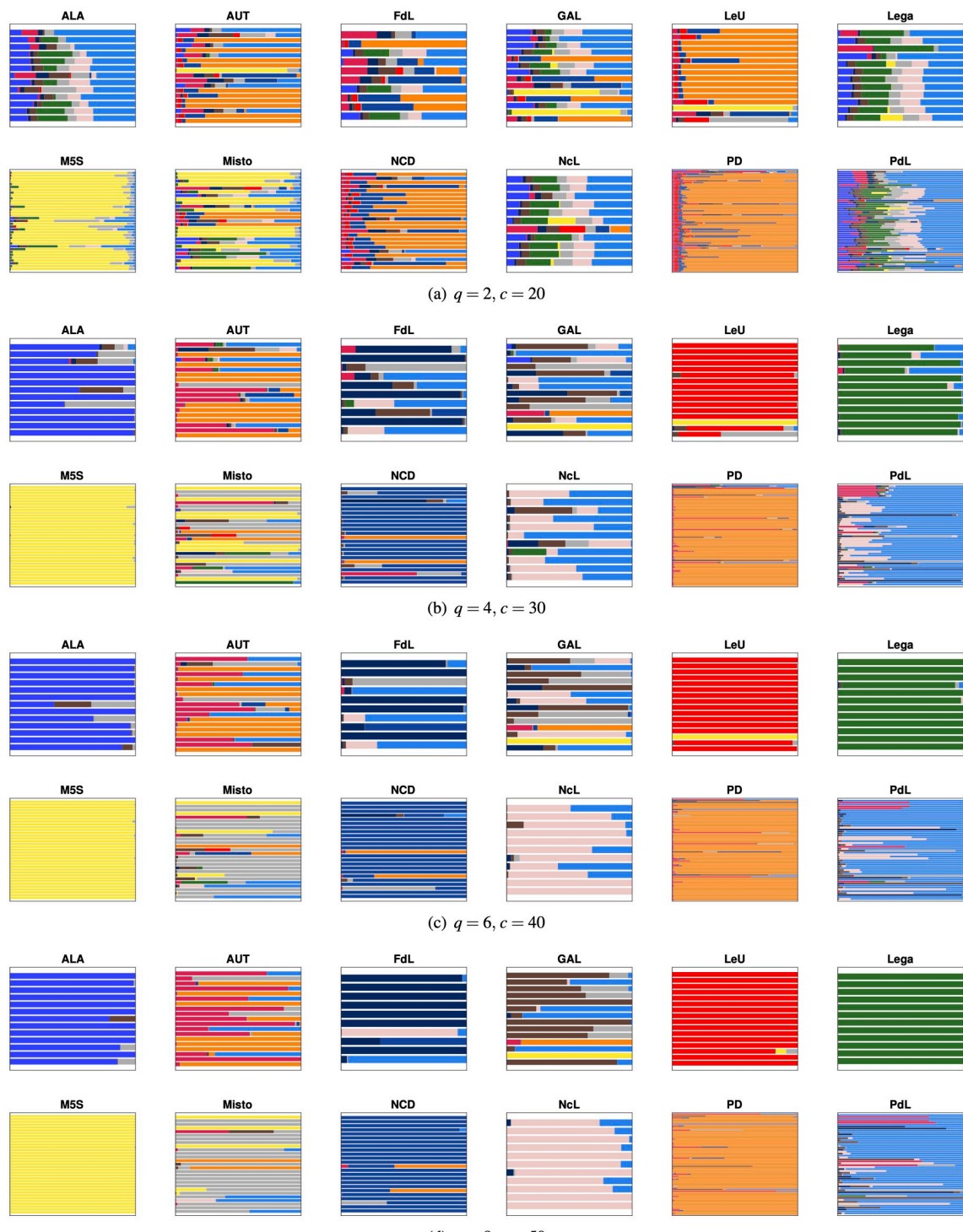

**Fig 7. Segmentation plots for different values of *q* and *c*.**

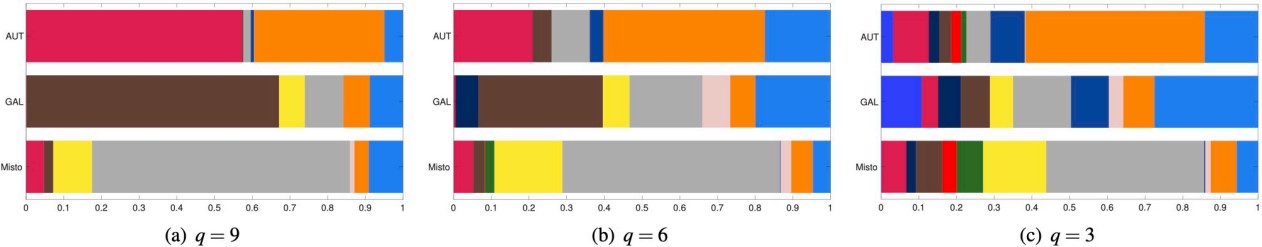

**Fig 8. Segmentation plot of the mean DNA of senators belonging to the groups "Misto", "GAL" and "AUT" for different values of $q$ and $c = 160$ (corresponding to non-sparse PCA).**

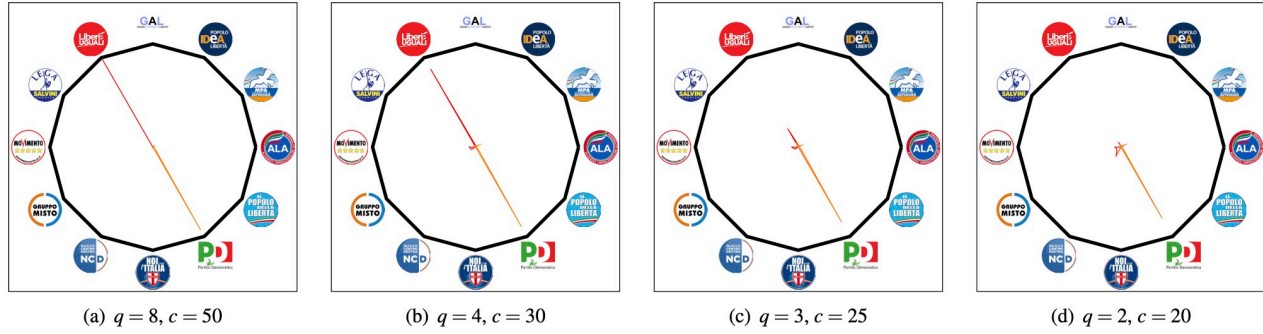

**Fig 9. Political affinity maps of the mean DNA of senators belonging to "LeU" and "PD' groups for different values of $q$ and $c$.**

elected candidates in this party to pay a penalty fine if the official party voting guidelines are not followed;

- for small values of $q$ the senators belonging to "LeU" start to shift towards the group "PD": this is coherent since both groups share a common political orientation (they are both left parties) and the "LeU" group foundation is linked to an internal split of the "PD" group; see Fig 9.

## Concluding remarks

In this paper, we presented a numerical technique that, based on publicly available voting data, generates explanatory maps of hidden interconnections among voters nominally belonging to a given number of political or ideological groups. The proposed method is based on a Gaussian mixture generative model that we use as a prior to compute a voter's posterior influences (the Political DNA), given evidence of his/her votes; see Fig 2. We applied this approach to a data set pertaining to the votes of 335 members of the Italian Senate on 160 bills during the XVII Legislature. Although the analysis has been carried out by clustering senators according to their nominal membership, other approaches are possible. For instance, Fig 10 depicts the DNA obtained by clustering senators either based on their general political orientation (Center-Left-Right) or on their role in the parliament (Government-Opposition), thus highlighting the flexibility of the proposed numerical technique. Further, while the DNA approach is here presented in a political context, we believe that the kind of interpretability it offers makes it suitable to broader application endeavors, such as in the analysis of behaviors, influence and preferences in markets, advertisement, or other social interaction contexts that are based on votes, preferences, purchases, etc.

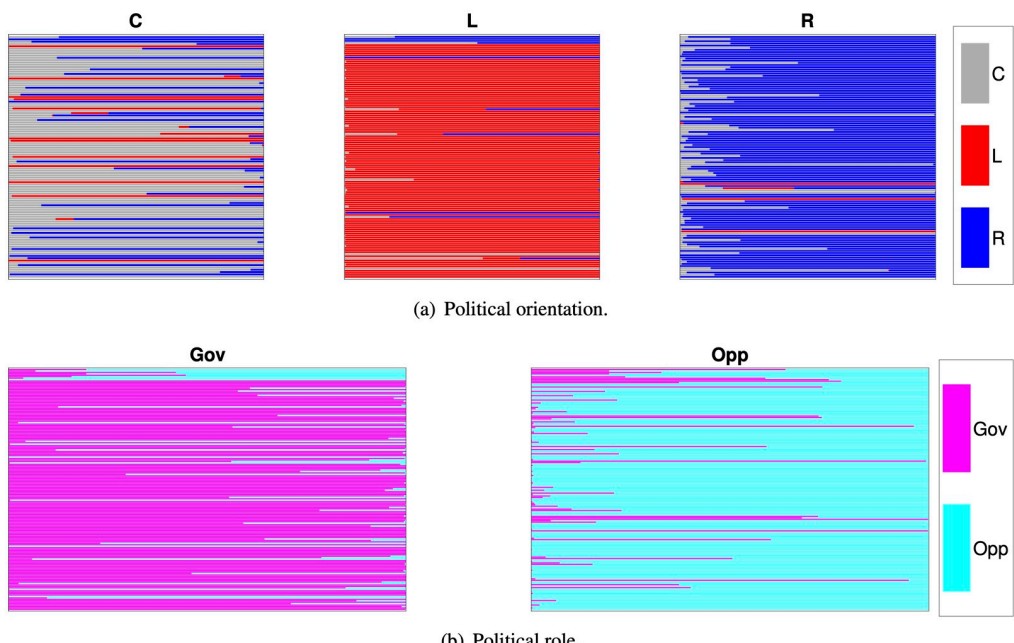

(a) Political orientation.

(b) Political role.

**Fig 10. Segmentation plot obtained by clustering senators on the basis of their orientation and their role ($q = 6$, $c = 30$).**

## Acknowledgments

The authors are indebted to Vincenzo Smaldore and to OpenPolis Foundation for providing data and insights on the interpretation of the results. The authors would also like to thank Antonio Longo for a first analysis of the data, and Antonio Santangelo, Francesco Ruggiero and the Staff of Nexa Center for interesting conversations on topics related to this paper.

## Author Contributions

**Conceptualization:** Corrado Possieri, Chiara Ravazzi, Fabrizio Dabbene, Giuseppe C. Calafiore.

**Methodology:** Corrado Possieri, Chiara Ravazzi, Fabrizio Dabbene, Giuseppe C. Calafiore.

**Software:** Corrado Possieri.

**Supervision:** Corrado Possieri, Chiara Ravazzi, Fabrizio Dabbene, Giuseppe C. Calafiore.

**Writing – original draft:** Corrado Possieri, Chiara Ravazzi.

**Writing – review & editing:** Corrado Possieri, Chiara Ravazzi, Fabrizio Dabbene, Giuseppe C. Calafiore.

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
