## [Decision Letter · Decision Letter 0]

27 Jul 2020

PONE-D-20-21006

A new metric for understanding hidden political influences from voting records

PLOS ONE

Dear Dr. Possieri,

Thank you for submitting your manuscript to PLOS ONE. After careful consideration, we feel that it has merit but does not fully meet PLOS ONE’s publication criteria as it currently stands. Therefore, we invite you to submit a revised version of the manuscript that addresses the points raised during the review process.

We look forward to receiving your revised manuscript.

Kind regards,

Gabriele Oliva, Ph.D

Academic Editor

PLOS ONE

Journal Requirements:

Additional Editor Comments (if provided):

Two reviews were collected both very positive, although both reviewers suggest minor improvements or request minor clarifications.

For this reason, I invite the authors to prepare a minor revision and to address such comments.

Reviewers' comments:

Reviewer's Responses to Questions

**Comments to the Author**

1. Is the manuscript technically sound, and do the data support the conclusions?

Reviewer #1: Yes

Reviewer #2: Yes

2. Has the statistical analysis been performed appropriately and rigorously? 

Reviewer #1: Yes

Reviewer #2: Yes

3. Have the authors made all data underlying the findings in their manuscript fully available?

Reviewer #1: Yes

Reviewer #2: Yes

4. Is the manuscript presented in an intelligible fashion and written in standard English?

Reviewer #1: Yes

Reviewer #2: Yes

5. Review Comments to the Author

Reviewer #1: The authors must be complimented for a very interesting and novel application of a classical (but probably still the more straightforward and meaningful) technique as PCA. The strategy adopted by the authors allows for a consistent analysis of both q and c space giving a clear picture of both senator profile and bill relevance together with an appreciation of the the 'internal homogeneity' of the different political parties.

I have only one (very minor) request of clarification that I think could be useful to the readers to fully appreciate the proposed methodology. The authors affirm: ...Therefore, noise filtering through decreasing the values of q and c reveals subtle structures of the political affinities among voters...;

This statement is strictly dependent from the use of SPCA instead of 'plain' PCA in which happens exactly the opposite ('subtle structures' are confined in minor components and filtering them out only maintains the 'global trends' see for example: Roden, J. C., King, B. W., Trout, D., Mortazavi, A., Wold, B. J., & Hart, C. E. (2006). Mining gene expression data by interpreting principal components. BMC bioinformatics, 7(1), 194., Censi, F., Calcagnini, G., Bartolini, P., & Giuliani, A. (2010). A systems biology strategy on differential gene expression data discloses some biological features of atrial fibrillation. PLoS One, 5(10), e13668., Giuliani, A., Colosimo, A., Benigni, R., & Zbilut, J. P. (1998). On the constructive role of noise in spatial systems. Physics Letters A, 247(1-2), 47-52.).

In any case this is only a very minor remark and the manuscript can be even accepted with no modifications at all.

Reviewer #2: The manuscript is very interesting and sound, I have few minor comments for the authors.

Reading the introduction of the paper I wasn't 100% sure that the bills used to conduct the analysis were those in which the vote is not secret. Even if it would be impossible to perform the analysis using secret votes I think that the Introduction should clearly state that only "public" bills were used and that those bills are just a part of the total ensemble.

The explanation of the visualization proposed in Figure 3 (that according to the authors is part of the novelty of the paper) is very brief and somewhat unclear. I understand the coordinates of the party and I also understand the presence of spikes proportional to \\pi in the direction of the party. I don't understand why some shapes are resulting from the Figure, are those the result of lines going between adjacent and non-zero \\beta values? If this is the case, what is the interpretation of such shapes? Plus, how this plot is different with respect to a radar plot?

This drawbacks also affect Figure 5.

Finally, Figure 3 and Figure 4 are basically two visualizations for the same thing. Similarities, differences and potential use of both (separately and combined) could be discussed in more detail.

6. PLOS authors have the option to publish the peer review history of their article (what does this mean?). If published, this will include your full peer review and any attached files.

Reviewer #1: **Yes: **Alessandro Giuliani

Reviewer #2: No

---

## [Author Response · Author response to Decision Letter 0]

30 Jul 2020

Response to the Editor

We would like to thank the Editor for the constructive reviewing process that helped us improve the quality of the manuscript and for the supportive and motivating comments about our work. We have carefully considered all the comments from the Reviewers, and their suggestions have been incorporated into the revised version of the paper, as detailed below.

Response to Reviewer#1

We thank the Reviewer for his appreciation of our the paper, for the motivating and supportive comments, and for the useful suggestions that helped us preparing this revised version of our paper.

The point raised by the Reviewer is very interesting and therefore we added a detailed explanation following his suggestion. Namely, we added the suggested references and following paragraph to the discussion at page 13 of the revised version of the manuscript:

“It is worth noticing that this filtering property is the result of the usage of SPCA for dimensionality reduction. In fact, subtle structures are usually confined in minor components [29]-[31], and filtering them out usually only maintains the global trends. On the other hand, SPCA allows one to focus just on the most relevant votes, hence undercovering hidden influences among individuals with different ideologies.”

Response to Reviewer#2

We thank the Reviewer for his/her appreciation of the paper, for the supportive comment, and for the useful suggestions that helped us preparing this revised version.

We revised the introduction so to clarify the class of votes that have been taken into account in performing the analysis of the Italian Senate during the XVII Legislature. Namely, we stressed the fact that our focus are voting data that are classified by OpenPolis as key votes, i.e., those votes that are publicly available (non secret) and considered as the most important, both for the relevance of the subject matter and for the political value. We also acquired the nominal membership of each senator to her/his political group, which will be used as side information. We further stressed that these votes constitute just a portion of the total ensemble of votes made by a senator, which also include, e.g., the ones made in secret ballots.

In the revised version of the paper, we better acknowledged that the Political Affinity Map is essentially a spider chart (usually referred to also as radar plot or Kiviat diagram) in which the length of each “spike” is proportional to the influence of the yth group to the ith senator and adjacent “spikes” are connected via a segment. Depending on the ordering of the political groups, this type of plots allows one to undercover the shift of each senator toward political parties different from the nominal one. For instance, larger areas of the polytope indicate that the ideology of the senator is influenced by several political parties, whereas unitary spikes in the direction of his/her nominal affiliation indicate that his/her ideology is consistent with the one of the party. Further, the Political Affinity Map allows one to determine senators with similar ideologies, political clusters, and the presence of outliers; see, e.g., [28].

In order to better discuss the relation between Political Affinity Maps and Segmentation Plots, we added the following paragraph at page 13 of the revised version of our paper:

“It is worth noting that the Political Affinity Map and the Segmentation Plot are just two different graphical representations of the Political DNA. These two representations allows to gather different information about the ideology of senators. For instance, as already pointed out above, the Political Affinity Map is very useful to identify clusters and to detect the presence of outliers. On the other hand, the Segmentation Plot is very useful to undercover the reciprocal influence among different parties and to identify the leading groups. Therefore, the combined use of these two representation allows one to have a complete picture of the relationships among different parties.”

---

## [Decision Letter · Decision Letter 1]

18 Aug 2020

A new metric for understanding hidden political influences from voting records

PONE-D-20-21006R1

Dear Dr. Possieri,

We’re pleased to inform you that your manuscript has been judged scientifically suitable for publication and will be formally accepted for publication once it meets all outstanding technical requirements.

Kind regards,

Gabriele Oliva, Ph.D

Academic Editor

PLOS ONE

Additional Editor Comments (optional):

Both reviewers are satisfied by the review, and I concur with their evaluation.

Reviewers' comments:

Reviewer's Responses to Questions

**Comments to the Author**

1. If the authors have adequately addressed your comments raised in a previous round of review and you feel that this manuscript is now acceptable for publication, you may indicate that here to bypass the “Comments to the Author” section, enter your conflict of interest statement in the “Confidential to Editor” section, and submit your "Accept" recommendation.

Reviewer #1: All comments have been addressed

Reviewer #2: (No Response)

2. Is the manuscript technically sound, and do the data support the conclusions?

Reviewer #1: Yes

Reviewer #2: (No Response)

3. Has the statistical analysis been performed appropriately and rigorously? 

Reviewer #1: Yes

Reviewer #2: (No Response)

4. Have the authors made all data underlying the findings in their manuscript fully available?

Reviewer #1: Yes

Reviewer #2: (No Response)

5. Is the manuscript presented in an intelligible fashion and written in standard English?

Reviewer #1: Yes

Reviewer #2: (No Response)

6. Review Comments to the Author

Reviewer #1: The authors commented on the issue I suggested them. In any case this is an excellent paper and the authors must be complimented for their sensible use of data analysis tools.

Reviewer #2: The authors reviewed the manuscript fixing all the minor issues. I recommend the paper for publication for publication in PLOS ONE

7. PLOS authors have the option to publish the peer review history of their article (what does this mean?). If published, this will include your full peer review and any attached files.

Reviewer #1: **Yes: **Alessandro Giuliani, Istituto Superiore di Sanità, Roma, Italy

Reviewer #2: No

---

## [Editor Report · Acceptance letter]

20 Aug 2020

PONE-D-20-21006R1 

A new metric for understanding hidden political influences from voting records 

Dear Dr. Possieri:

I'm pleased to inform you that your manuscript has been deemed suitable for publication in PLOS ONE. Congratulations! Your manuscript is now with our production department. 

Kind regards, 

on behalf of

Dr. Gabriele Oliva 

Academic Editor

PLOS ONE